# A Comparative Analysis of the Historical Accuracy of the Point Precipitation Frequency Estimates of Four Data Sets and Their Projections for the Northeastern United States

**Shu Wu [1,\*]** , **Momcilo Markus [2]**, **David Lorenz [1]**, **James R. Angel [2]** and **Kevin Grady [2]**

[1]  Nelson Institute Center for Climatic Research, University of Wisconsin–Madison, Madison, WI 53706, USA; david.lorenz@wisc.edu

[2]  Prairie Research Institute, University of Illinois at Urbana-Champaign, Champaign, IL 61801, USA; mmarkus@illinois.edu (M.M.); jimangel@illinois.edu (J.R.A.); kagrady2@illinois.edu (K.G.)

\*  Correspondence: swu33@wisc.edu; Tel.: +1-608-209-8688

**Abstract:** Many studies have projected that as the climate changes, the magnitudes of extreme precipitation events in the Northeastern United States are likely to continue increasing, regardless of the emission scenario. To examine this issue, we analyzed observed and modeled daily precipitation frequency (PF) estimates in the Northeastern US on the rain gauge station scale based on both annual maximum series (AMS) and partial duration series (PDS) methods. We employed four Coupled Model Intercomparison Project Phase 5 (CMIP5) downscaled data sets, including a probabilistic statistically downscaled data set developed specifically for this study. The ability of these four data sets to reproduce the observed features of historical point PF estimates was compared, and the two with the best historical accuracy, including the newly developed probabilistic data set, were selected to produce projected PF estimates under two CMIP5-based emission scenarios, namely Representative Concentration Pathway 4.5 (RCP4.5) and Representative Concentration Pathway 8.5 (RCP8.5). These projections indeed demonstrate a likely increase in PF estimates in the Northeastern US with noted differences in magnitudes and spatial distributions between the two data sets and between the two scenarios. We also quantified how the exceedance probabilities of the historical PF estimate values are likely to increase under each scenario using the two best performing data sets. Notably, an event with a current exceedance probability of 0.01 (a 100-year event) may have an exceedance probability for the second half of the 21st century of ≈0.04 (a 27-year event) under the RCP4.5 scenario and ≈0.05 (a 19-year event) under RCP8.5. Knowledge about the projected changes to the magnitude and frequency of heavy precipitation in this region will be relevant for the socio-economic and environmental evaluation of future infrastructure projects and will allow for better management and planning decisions.

**Keywords:** frequency estimates; downscaling; future projections; RCP4.5; RCP8.5; changing of exceedance; Northeastern US

---

## 1. Introduction

The time series of extreme precipitation events in the United States are not stationary [1], as the statistics describing them have been observed to change over the past several decades [2]. They will likely continue to change in the future as the climate changes (e.g., [3]). In the Northeastern US, in particular, extreme precipitation events have increased the most dramatically compared with any other region in the United States in the past several decades [4–7]. These changes then need to

be investigated so that design standards can be met when constructing infrastructure that requires information about precipitation extremes. Many metrics could be used to describe the features of these extreme precipitation events, among which precipitation frequency (PF) analysis was selected for this study, as it is particularly important to hydrology design [8]. The goal of PF analysis is to obtain a useful estimate of a quantile for a relevant return period [9,10].

There are several methods of estimating future PFs. One method is to fit a nonstationary PF model to the observational data, such as a generalized extreme value (GEV) distribution, whose location, scale, and shape parameters are functions of time [11,12]. Consequently, the PF estimates are also functions of time with this method; substituting future dates into these PF functions results in future PF estimates. This method uses only historical observation data to obtain future PF estimates, so it may also be referred to as an observation extrapolation method. However, a key uncertainty of this method is that the linear trend assumption [12] may not hold in the future.

Another method is to use the output of climate models under different emission scenarios [2,13,14]. Climate models have demonstrated the ability to reproduce many observational features [15]; however, these models still present many uncertainties [16]. One of these uncertainties is caused by their focus mainly on larger climatic features on the continental or global scale. However, many hydrologic applications require smaller scales for which comparing model output with observations often involves a so-called downscaling process since the model resolutions are too coarse to be compared directly with the finer station observations. This downscaling process from the large to local scale, either done statistically [17,18] or dynamically [19,20], adds another layer of uncertainty when interpreting the results.

In this paper, we use model output to analyze point PF estimates for a large number of individual rain gauge stations throughout the Northeastern US. We compute PF estimates at each station for the historical period of 1960–2005 using each of the four downscaled Coupled Model Intercomparison Project Phase 5 (CMIP5) precipitation data sets, including three commonly used ones, and a fourth probabilistic data set developed for this study (Table 1). These modeled PF estimates are then compared with the observational PF estimates and each other. Models that are able to better reproduce historical data are assumed to be more accurate when making future projections [2,21,22]. The goal is then to find the data sets with the best overall historical accuracies to produce projected PF estimates and to quantify how the exceedance probabilities of the historical PF estimates are likely to change in the future under certain emission scenarios. These results will examine and highlight the need to reevaluate the existing PF estimates in the light of climate change.

This paper is organized as follows. We introduce the data sets and methodologies we will employ in Section 2, followed by comparisons between the observed and downscaled model data for the historical period in Section 3. Informed by these results, we select the best data sets and use them to present possible changes to the PF estimates in the future under two different Representative Concentration Pathways (RCPs), namely RCP4.5 and RCP8.5, in Section 4. The results in Section 4 demonstrate that PF estimates are very likely to increase in the future, suggesting that the exceedance probabilities of the historical PF estimates are also likely to increase. These changes are discussed in Section 5. Section 6 summarizes the results and provides conclusions and further discussions.

**Table 1.** **Intergovernmental Panel on Climate Change** (IPCC) models used to generate the downscale data sets.

| Data Set | Models Used |
| --- | --- |
| UWPD (24) | ACCESS1-0; ACCESS1-3; CanESM2; CMCC-CESM; CMCC-CM CMCC-CMS; CNRM-CM5; CSIRO-Mk3-6-0; GFDL-CM3; GFDL-ESM2G GFDL-ESM2M; HadGEM2-CC; inmcm4; IPSL-CM5A-LR; IPSL-CM5A-MR IPSL-CM5B-LR; MIROC5; MIROC-ESM; MIROC-ESM-CHEM; MPI-ESM-LR MPI-ESM-MR; MRI-CGCM3; MRI-ESM1; NorESM1-M |

**Table 1.** *Cont.*

| Data Set | Models Used |
| --- | --- |
| LOCA (32) | ACCESS1-0; ACCESS1-3; bcc-csm1-1; bcc-csm1-1-m; CanESM2 CCSM4; CESM1-BGC; CESM1-CAM5; CMCC-CM; CMCC-CMS CNRM-CM5; CSIRO-Mk3-6-0; EC-EARTH; FGOALS-g2; GFDL-CM3 GFDL-ESM2G; GFDL-ESM2M; GISS-E2-H; GISS-E2-R; HadGEM2-AO HadGEM2-CC; HadGEM2-ES; inmcm4; IPSL-CM5A-LR; IPSL-CM5A-MR MIROC5; MIROC-ESM; MIROC-ESM-CHEM; MPI-ESM-LR; MPI-ESM-MR MRI-CGCM3; NorESM1-M |
| BCCAv2 (20) | ACCESS1-0; bcc-csm1-1; CanESM2; CCSM4; CESM1-BGC CNRM-CM5; CSIRO-Mk3-6-0; GFDL-CM3; GFDL-ESM2G; GFDL-ESM2M inmcm4; IPSL-CM5A-LR; IPSL-CM5A-MR; MIROC5; MIROC-ESM; MIROC-ESM-CHEM; MPI-ESM-LR; MPI-ESM-MR; MRI-CGCM3; NorESM1-M |
| NA-CORDEX (6) | CanESM2: {CRCM5(0.44°); RCA4(0.44°); CanRCM4(0.22°, 0.44°)}; EC-EARTH: {HIRHAMS(0.44°); RCA4(0.44°)} |

Note: For the full model names, we refer to the CMIP5 website: https://cmip.llnl.gov/cmip5/docs/CMIP5_modeling_groups.docx.

## 2. Data and Methodologies

### 2.1. The Observational and Downscaled Model Data Sets

We examined observational data collected by the National Oceanic and Atmospheric Administration (NOAA) and used in the development of Atlas 14 [23]. The original data set included observations from 1218 rain gauge stations in the Northeastern US, of which 753 stations had a temporal coverage of over 80% for the period of 1960–2005 and were thus used in this study (Figure 1). Due to the limitations on the scope of this paper, we refer the reader to the published report [23] for more detailed quality control information about these data. The 1960–2005 period was selected for our analysis in order to have consistent comparisons with climate model outputs.

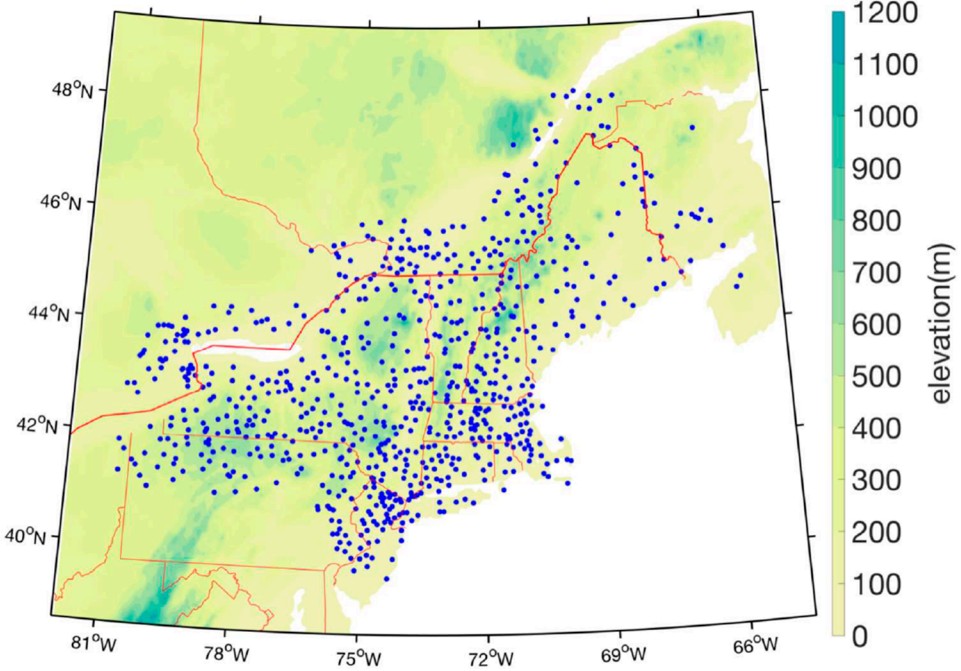

**Figure 1.** A map of the 753 selected stations in the Northeastern US. Shading represents elevation.

A large fraction of the study area lies along the northern Appalachian Mountains (Figure 1), which separate the coastal plain from the central lowlands. Local maxima of annual precipitation occur along

the coast and higher elevations while local minima occur along the northern and western interior lowlands [4].

In addition to this observational data, four downscaled CMIP5 model data sets (Table 1) were analyzed. The first data set is a newly developed statistically downscaled product based on CMIP5 model output, hereafter referred to as the UWPD (University of Wisconsin Probabilistic Downscaling) data set [17]. This downscaled data set covers the Central and Eastern United States and Southern Canada at a $0.1 \times 0.1$ degree resolution. The downscaling methodology is fundamentally probabilistic [17]: The large-scale variables in a climate model do not determine the precise values of the downscaled variables. Instead, the large scale determines the likelihood of potential values of the downscaled variables; i.e., the large-scale variables determine the probability density function (PDF) of the downscaled variables. In the UWPD data, parametric PDFs are used, where the parameters of the distribution depend on the large-scale predictors. The simplest example of this approach is the use of ordinary least squares regression (OLS) to predict y from x. If the assumptions of OLS hold, then the expected value of y is Gaussian with a mean of the form ax + b (a and b are constants) and constant variance. In the UWPD data, the precipitation downscaling is divided into two components: (1) Estimate the probability that a day has precipitation or not (i.e., a binary variable), and (2) estimate the precipitation amount if the day is "wet." Logistic regression is used for the precipitation occurrence (first component), while a generalized gamma distribution [24] with parameters conditioned on the large scale is used for the precipitation amount (second component). Logistic regression was used because it is the appropriate regression analysis for binary data. Several potential link functions were explored for logistic regression, including the probit and several skewed link functions. It was found that the standard logit link function fit the observed data best. For the precipitation amount, the exponential, gamma, Weibull, and generalized exponential distributions were insufficiently flexible to capture the precipitation extremes. Therefore, the generalized gamma distribution, which has two shape parameters instead of one or zero, was used. The generalized gamma distribution includes the gamma and the Weibull distribution as special cases. Note, a single generalized gamma distribution is not used for all days, instead the parameters of the generalized gamma distribution depend on the daily varying large-scale atmospheric conditions. The precipitation amount methodology is very similar to that of the gamma distribution approach that Kirchmeier et al. [25] used for wind speed. The station data used to train the statistical models consist of daily precipitation accumulation from the National Weather Service (NWS)'s Cooperative Observer Program (1950–2009) and Environment Canada's Canadian Daily Climate Data (1950–2007) for approximately 4000 stations in the United States and Canada. Objective methods are applied to correct the hour of observation and remove stations with large biases in wet-day frequency [26], which results in the removal of 21% of stations. Only stations with at least 30 years of data and fewer than five missing days per month are included.

The large-scale atmospheric state for training the statistical models was obtained from the National Centers for Environmental Prediction (NCEP)–National Center for Atmospheric Research (NCAR) reanalysis [27]. The large-scale predictors include modeled precipitation itself, as well as variables involving moisture, wind, and static stability [17]. The large-scale fields were incorporated as predictors using the methodology in Kirchmeier et al. [25]. All statistical fits were cross-validated and only predictors with skill on independent data were used. To account for seasonality [4], separate fits were performed for each calendar month.

The time varying PDF parameters at each station that result from the above methodology were then interpolated in space [17]. The PDF parameters vary relatively slowly in space because they are determined from the large-scale atmospheric state; therefore, the interpolation used here results in almost no change in the statistics of the extreme events. Interpolating precipitation values, on the other hand, results in large reductions in precipitation variances due to the small-scale variability of precipitation. To generate "standard precipitation data" from the PDFs, random numbers are drawn from the PDFs to generate a "realization" of the small scale given the large scale. Such realizations are obviously not unique and are instead infinite in number and all equally likely. The random nature

of the data accounts for the fact that for each particular large-scale state, there are multiple possible evolutions of the small-scale atmosphere that are consistent with this large-scale state. For example, the large-scale atmosphere on a particular day may be favorable for the development of small-scale thundershowers, but any individual point may have only a 50% probability of actually being hit by one of these storms. When generating realizations, the random numbers are correlated in space and time so that the spatial and temporal correlations of the downscaled variables are similar to observations. This is achieved by forcing an advection-diffusion model with independent random numbers and then using the output of the advection-diffusion model for drawing realizations from the PDFs. The winds at 3000 m altitude are used for the horizontal winds in the advection-diffusion model. The CMIP3 version of these data has been shown to have good skill [14,17,18,28]. For further information about this data set, we refer the reader to the website [29].

The second data set is the Localized Constructed Analogs (LOCAs) statistically downscaled data set [30]. These LOCA data have been widely used in government reports, such as the US Global Change Research Program (USGCRP, [31]). We used the downscaled results from 32 models from this data set (Table 1), each with a 1/16-degree resolution ($\approx$6 km by 6 km). One simulation of each model was selected to form an ensemble of 32 members. For further information about this data set, we refer the reader to the LOCA website [32].

The third data set is the Bias-Correction and Constructed Analogs data (BCCAv2, [33]). The BCCAv2 data set has an ensemble size of 20 (one simulation of each model, Table 1) with 1/8-degree resolutions. For further information about this data set, we refer the reader to the website [34].

The final data set is the North America Coordinated Regional Downscaling Experiment data (NA-CORDEX, [35]). Although the previous three data sets were statistically downscaled (local variables are approximated from the large-scale predictor fields using statistical relationships), NA-CORDEX is instead dynamically downscaled. The data are generated by using the output of global models as lateral boundary and initial conditions to run high-resolution regional models. We selected downscaled data from two global models, namely CanESM2 and EC-EARTH, since the outputs of these two models are available across the historical, RCP4.5, and RCP8.5 periods. The CanESM2 model is paired with three regional models, namely CRCM5, RCA4, and CanRCM4, and EC-EARTH is paired with two regional models, namely RCA4 and HIRHAMS. All the data have a resolution of 0.44 degrees with the CanESM2 and CanRCM4 pairing having an extra simulation at a resolution of 0.22 degrees. This results in an ensemble of six members (Table 1). For further information about this data set, we refer the reader to the website [36].

The NA-CORDEX data set has not been completely generated yet, and our results (see Section 3.2) show that the BCCAv2 data set has relatively larger biases compared to the other data sets. Thus, our analysis will be mainly focused on the UWPD and LOCA data. However, the BCCAv2 and NA-CORDEX data also show some potential to perform well, especially the NA-CORDEX data; therefore, we provided some comparison results for these data as well for completeness. It should also be noted that the BCCAv2 and NA-CORDEX data are relatively coarse in resolution. This may lead to larger representation errors when comparing with point precipitation observations. Therefore, we applied an inverse of the empirical area reduction factors [14,37], producing 1.05 for the BCCAv2 data, 1.09 for the 0.22 degrees NA-CORDEX data, and 1.16 for the 0.44 degrees NA-CORDEX data.

### 2.2. Methodologies

Similar to NOAA Atlas 14 [23,38], our paper presents PF estimate results based on the annual maximum series (AMS) and partial duration series (PDS) of daily precipitation data. AMS and PDS are both widely used to carry out PF analysis in hydrology [39]. AMS are generated based on the maximum daily rainfall of each year, so the typical length of an AMS equals the number of years in the record. All of the values except for the one largest in a year are eliminated in calculating the PF estimates, which could potentially exclude some large events if two or more occur in a given year. In contrast, PDS are generated by retaining the values over a preselected threshold [40]. Therefore, the

length of a PDS series is determined by the threshold, which may vary dramatically across different stations. Generally, the threshold is selected such that, on average, several events (2–4) occur per year, meaning the lengths of PDS are usually several times larger than those of the AMS for the same station and employ more observational data.

Because PDS use more data, one could presume that they give more significant PF estimates than AMS. However, more non-extreme values may be retained with PDS, which could lead to an underestimation of the final PFs, since PF analysis intrinsically focuses on extreme events. In practice, many products, such as NOAA Atlas 14, include results from both methods because AMS and PDS complement each other well and including both can also give more information on the uncertainties of the PF estimates. We too compared PF estimates from both AMS and PDS in our analysis to get a more complete understanding of how the downscaled data are able to simulate the observed features.

Calculating PF estimates using AMS versus PDS can be quite different in practice. For example, AMS are often associated with the generalized extreme value (GEV) distribution, while PDS are instead often associated with the generalized Pareto distribution (GPD) [40]. To make a direct comparison of AMS- and PDS-based PF results, we need to use consistent return periods for the two methods. However, the AMS return periods often need to be modified in order to do so since the AMS method tends to underestimate PF estimates for shorter return periods when compared with the PDS method [8]. For a given PDS return period, the adjusted AMS return period can be calculated as [41]:

$$\mathrm{T(AMS)} \,=\, \left(1 - \mathrm{e}^{-\frac{1}{\mathrm{T(PDS)}}}\right)^{-1}. \tag{1}$$

For example, a return period of 2 years for a PDS may be equated to a return period of 2.54 years for an AMS. In this paper, we used "T = [2 5 10 25 50 100 200 500 1000]" years for the PDS return periods and "T = [2.54 5.52 10.51 25.50 50.50 100 200 500 1000]" years for the corresponding AMS return periods so that our PDS- and AMS-based PF estimates could be compared directly.

A more proper way to describe PF estimates associated with AMS might be to use an annual exceedance probability (AEP) instead of return periods [23]. Assuming the relationship holds that the AEP is the reciprocal of the return period for an AMS, we still used return periods for the convenience of comparison in this paper. For more about exceedance probabilities, see Section 5.

To generate a PDS, a threshold needs to be determined first. Although some methods are used to select the PDS threshold objectively, such as mean residual life plots [40], checking these plots for all stations to select the proper thresholds is virtually impossible. As a compromise, we examined three different methods of selecting thresholds in order to cover the associated uncertainties. For the first method, we used an iteration scheme to select a PDS threshold such that each year, on average, has two exceedance events. We treated consecutive exceedance days as one event, retaining only the largest value of the event. The second method is similar to the first, but the PDS threshold was lowered such that there are three events per year on average. For the third method, we used the 98% quantile level of non-zero precipitation values as the PDS threshold. In this paper, these three methods are labeled as PDS2, PDS3, and PDS qt98. The resulting thresholds for each station are shown in Figure 2a. This figure shows that, for most stations, PDS3 uses the lowest threshold and PDS2 uses the highest, with PDS qt98 falling in the middle. Different thresholds lead to different lengths of the base time series for PDS PF analysis (Figure 2b). As expected, the lengths of PDS2 and PDS3 are just double and triple, respectively, the number of analysis years, or 46 in our case (1960–2005), while the length of the PDS qt98 series varies by station. Please note that the stations are ordered by their location along the *x*-axis in this figure. The station order appears to be relatively arbitrary but has some tendency to cluster adjacent stations together. However, these patterns were not examined in this study.

Once we obtained the AMS and PDS for each station, GEV and GPD distributions, respectively, were fitted to the data to get the relevant estimated parameters by using the maximum likelihood method. PF estimates were then calculated using exceedance probabilities ([40], also see Section 5 of this paper).

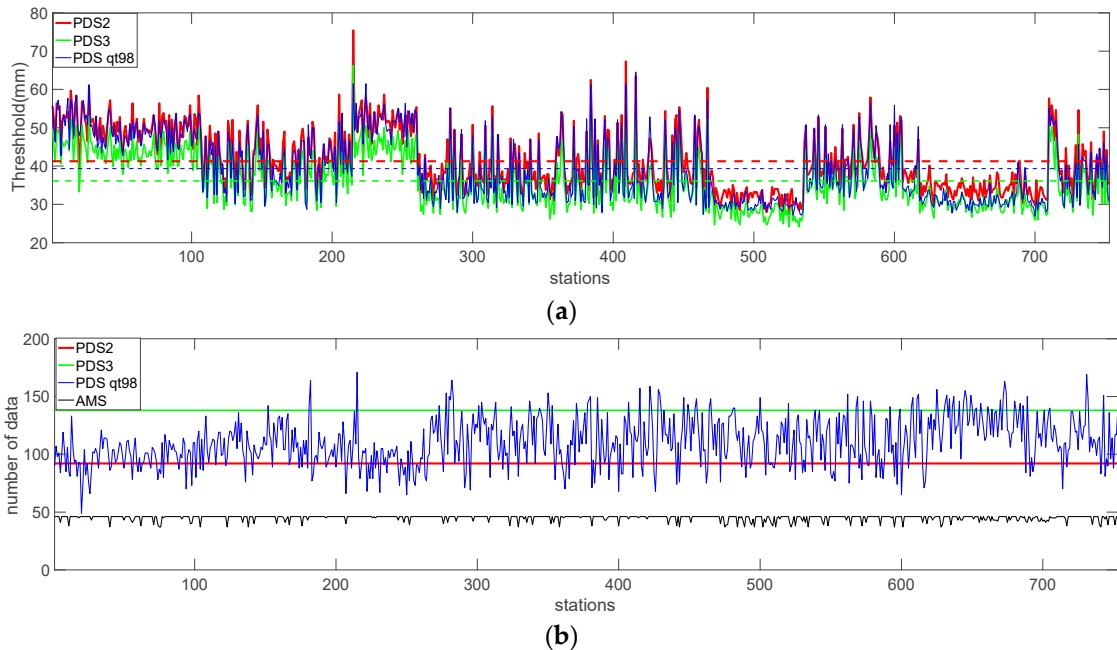

**Figure 2.** PDS thresholds (**a**) and length of data (**b**) used for PF analysis for each station. Red represents the results for PDS2, green for PDS3, blue for PDS qt98, and black for AMS.

## 3. Evaluations of the Downscaled Model Data for the Historical Period of 1960–2005

In this section, we evaluate the performance of the downscaled model data in terms of reproducing the observed features of the mean AMS, PDS, and PF estimates for different return periods. We will focus on the UWPD and LOCA data; some results for the BCCAv2 and NA-CORDEX data will also be provided at the end of this section. For this analysis, the closest model grid point to a rain gauge station was selected as its representation in the model.

### 3.1. Comparison of the Climatology of AMS and PDS of the Observed and Modeled Data

The 1960 to 2005 climatology (averages) of the observed AMS and PDS for each station are shown in the first row of Figure 3. The range is 40 to 120 mm, with larger values typically along the coast and smaller values inland. For the AMS (the first column of Figure 3), the differences between the UWPD data and these observed values are quite small in terms of difference ratios defined as (model − observation)/observation (Figure 3e). For over 80% of the stations, the difference ratio magnitudes are less than 10%, and there is no sign of systematic biases, as positive ratios are mixed together with negative ratios quite well. On the other hand, the LOCA data show systematic dry biases (negative ratios in Figure 3i) with difference ratio magnitudes of over 20% for most of the stations. For PDS, the climatology (Figure 3b–d) are in general smaller than for the AMS (Figure 3a), while the spatial distributions of the PDS are similar to each other. For every PDS threshold selection, the UWPD data generally have less than a 10% error compared with the observational results and no significant biases (Figure 3f–h), while the LOCA data generally have more than a 20% error and a systematic bias toward drier values, especially PDS qt98 (Figure 3j–l).

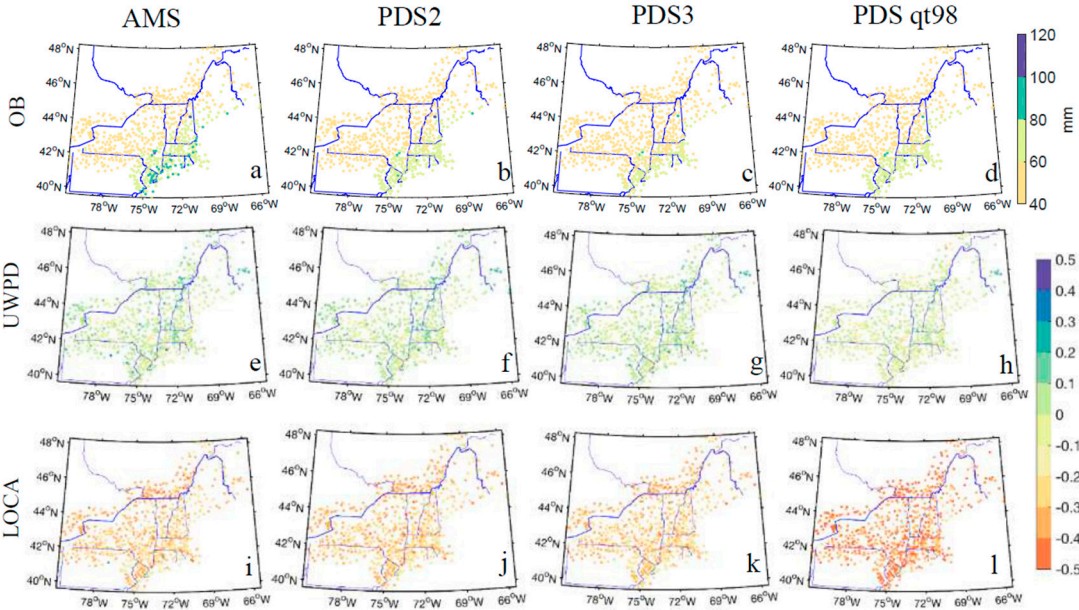

**Figure 3.** The 1960–2005 climatology of the AMS (**a**), PDS2 (**b**), PDS3 (**c**), and PDS qt98 (**d**), based on the observational data and the relative differences defined as (model – observation)/observation) for the UWPD (**e**–**h**) and LOCA (**i**–**l**) data.

### 3.2. Comparison of PF Estimates of the Observed and Modeled Data

In this section, we directly compare the observational PF estimates with those of the downscaled data to assess how well the model data may perform in estimating historical PF values. Figure 4 shows the PF estimate comparison results for the UWPD data, and Figure 5 shows the results for the LOCA data. The rows in each figure represent PF estimates for return periods of 2, 25, and 100 years, respectively. For each return period, we compared PF estimates from the downscaled data based on the AMS and each of the three PDS methods with their respective observed counterparts to get a more complete understanding. The *y*-axis represents the PF estimates calculated based on observations, and the *x*-axis represents the PF estimates based on the downscaled data. For each station, an ensemble mean (red dots) and standard deviation (blue bars) based on the PF estimates of the ensemble members of that downscaled data set can be calculated and used to estimate the mean value and associated uncertainty for that station. To assist in the comparison, lines of "y = x" are also plotted in red in each figure. More points above the line "y = x" mean the observed values are larger than the downscaled values, thus indicating dry biases in the models and vice versa.

Overall, the AMS and PDS demonstrate quite consistent features when compared. Results for the UWPD data (Figure 4) show that most of the points sit well around the line, "y = x", suggesting the UWPD PF estimates are close to the observed PF estimates, especially for shorter return periods. For a 2-year return period (Figure 4a–d), the UWPD PF estimates have a spatial correlation of 0.93 with the observations for all the AMS and PDS methods, suggesting that the spatial variations of the observational PF estimates are well represented by the UWPD data. The root mean square error (RMSE), which is around 4.5 mm on average, is also small compared to the observed values (40–90 mm). As the return periods become longer, the correlation coefficients decrease to ≈0.85 for the 25-year return period (Figure 4e–h) and ≈0.75 for the 100-year return period (Figure 4i–l). The RMSE increases to ≈13 mm for the 25-year return period and to ≈25 mm for the 100-year return period. The uncertainties (the length of the blue lines) also increase as the return periods become longer. It should be noted that for the extremely high and low PF estimates, the UWPD data indeed show some large errors manifested as dry biases for extremely high PF estimates and wet biases for extremely low PF estimates. Yet the number of these biased stations is small compared to the total number of stations, allowing us to conclude that the UWPD data overall capture the major features of the observed PF estimates. In

contrast, the LOCA data consistently underestimate the PF estimate values across all the return periods (Figure 5), thus showing the same dry bias seen in Figure 3. The spatial correlation coefficients are quite high, however, indicating that the LOCA data can also capture the observed spatial variations relatively well. Standard deviations of the LOCA data (the length of the blue lines) are also smaller than for the UWPD data.

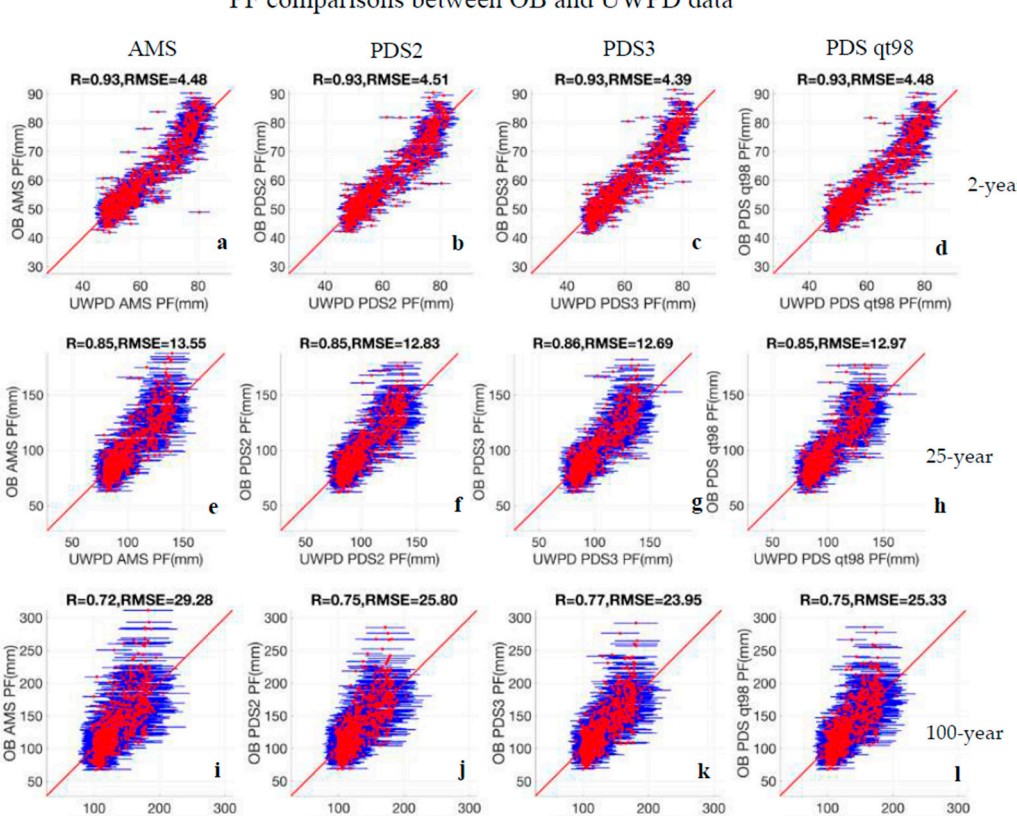

**Figure 4.** PF estimate comparisons between the observations and the UWPD data for return periods of 2 years (upper row), 25 years (middle row), and 100 years (lower row). The *y*-axis represents the PF estimates calculated based on observations, and the *x*-axis represents the PF estimates based on the downscaled data. For each station, an ensemble mean is represented by a red dot and the standard deviation is represented by a blue bar. Lines of "y = x" are also plotted for reference.

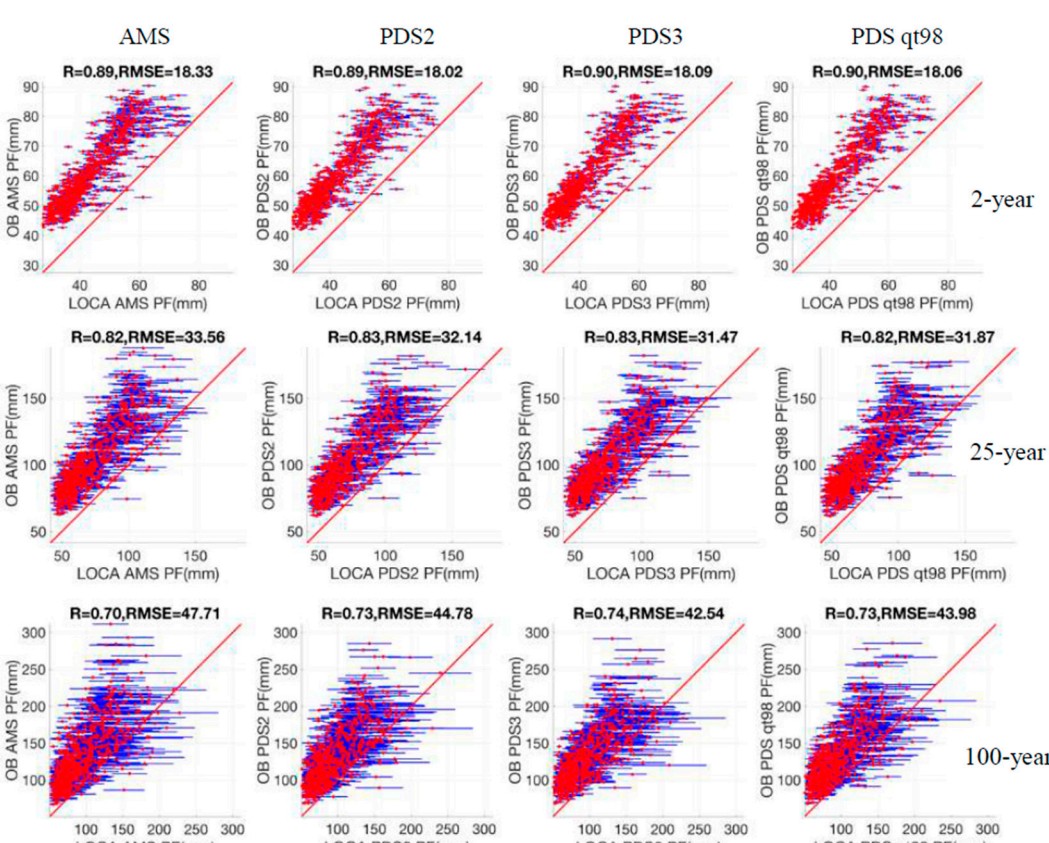

**Figure 5.** Similar to Figure 4 but for the LOCA data.

To summarize the comparison results, Figures 6 and 7 provide quantifications of skill, in terms of correlation coefficients and RMSE [8], respectively, for all four downscaled data sets across all return periods. The skills of the downscaled data sets decrease as return periods become longer. Quite consistently, we see that the UWPD data outperform the other data sets for all return periods. The LOCA data are the second best in terms of correlation skill, implying that although significant biases are present in the LOCA data, a simple bias correction may be adequate in making the LOCA data still useful for PF estimates. Interestingly, the only dynamically downscaled data set, NA-CORDEX, regardless of its relatively coarse resolution, also shows quite comparable skill for both the correlation coefficients and RMSE, although the uncertainties reflected by the ensemble spread are much larger than for all the statistically downscaled data sets. The RMSEs of the BCCAv2 data mainly reflect their systematic dry biases (figures not shown). Again, the different PF calculation methods based on the AMS and PDS show quite consistent results, regardless of the method used.

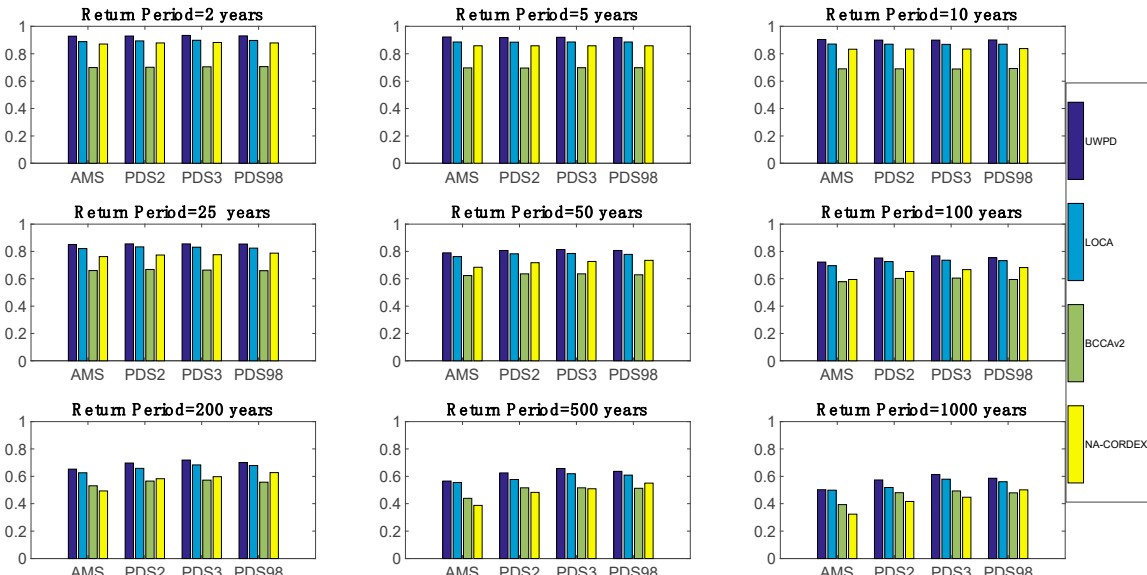

**Figure 6.** PF correlations between OB and downscaled data for different return periods. PF estimate correlation skills of the four downscaled data sets for different return periods: UWPD (dark blue), LOCA (light blue), BCCAv2 (light green), and NA-CORDEX (yellow).

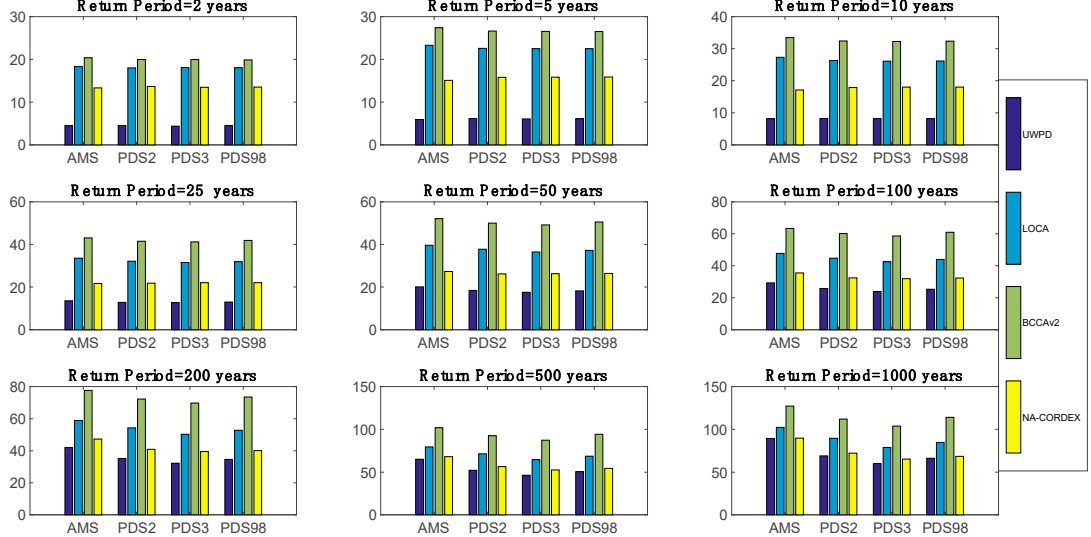

**Figure 7.** RMSE of PF estimates for downscaled data for different return periods. Similar to Figure 6 but for RMSE.

## 4. Projected PF Estimates

Based on our results in Section 3 in which the UWPD data were shown to capture major features of the observed PF estimates the best, we selected this data set for our primary analysis of future projections. Since the LOCA data set has been widely used by government reports and other publications [2], we also conducted parallel calculations based on this data set as well. Comparisons of the corresponding UWPD and LOCA results helps to determine how significant the findings really are. Doing so also gives some insight into other studies where the LOCA data have been used [4]. Our comparison study for the historical period in Section 3 showed that the PF estimates calculated by the different methods of AMS and PDS are quite similar. Therefore, for simplicity, we only show the PF estimate results that are based on the AMS for the following sections.

The projected data cover the period of 2006–2100. Knowing that a trend appears in this 95-year period, we divided the whole period into two smaller time periods: 2006–2053 (representing the first

half of the 21st century) and 2054–2100 (representing the second half). Within each time period, we assumed the PF estimates are stationary as we did for the historical period, although some trends may also be detectable within each period. Figures 8 and 9 show for the low emission (RCP4.5) and high emission (RCP8.5) scenarios, respectively, the relative PF difference ratio comparing the projected data with the historical period. The ratio is defined as (PF_future – PF_historical)/PF_historical. PF estimates for return periods of 25 years and 100 years are shown here as representative examples.

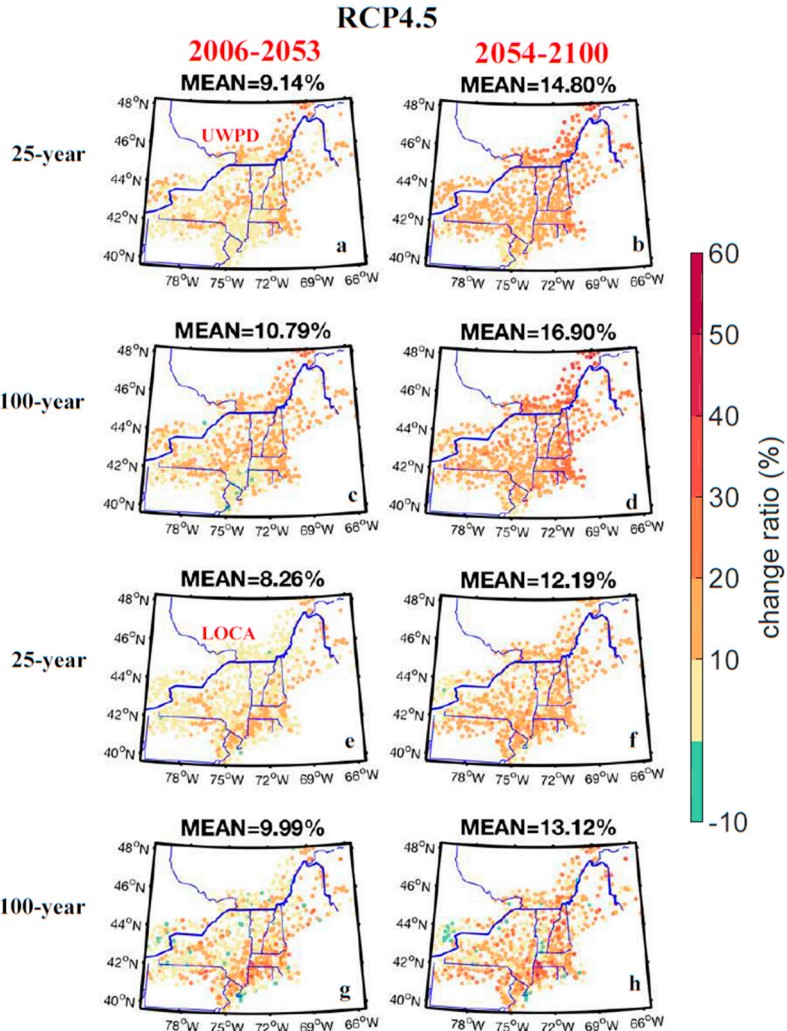

**Figure 8.** Relative PF difference ratios defined as (PF_future − PF_historical)/PF_historical for a 25-year return period (**a**,**b**,**e**,**f**) and a 100-year return period (**c**,**d**,**g**,**h**) under the RCP4.5 emission scenario.

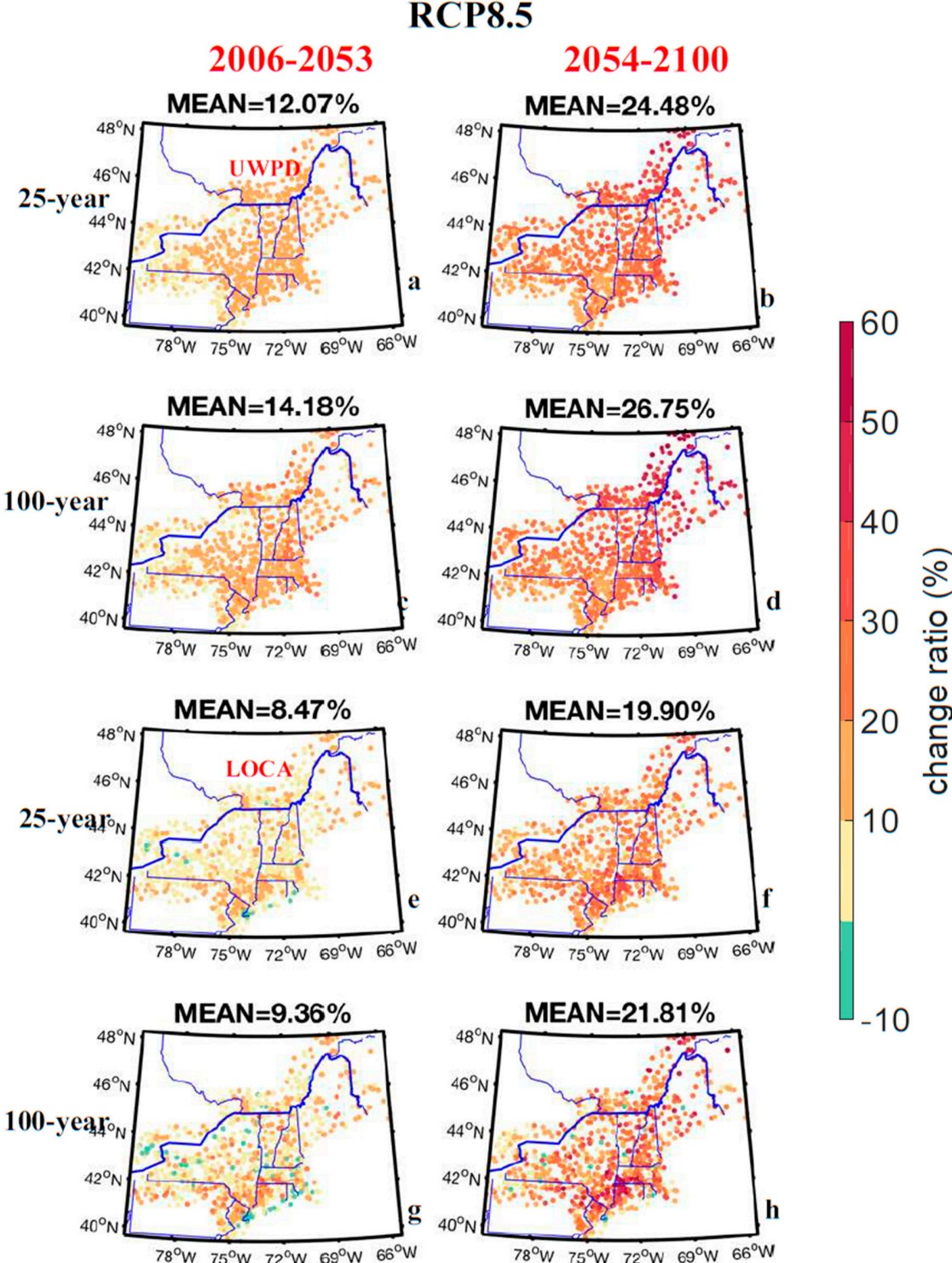

**Figure 9.** Relative PF difference ratios defined as (PF_future – PF_historical)/PF_historical for a 25-year return period (**a**,**b**,**e**,**f**) and a 100-year return period (**c**,**d**,**g**,**h**) under the RCP8.5 emission scenario.

Under the low emission scenario (RCP4.5) for the first half of the 21st century, the results based on the UWPD data show that, for a return period of 25 years (Figure 8a), PF estimates increase by a mean relative ratio of 0.09, which translates to a 9% increase. New Jersey and the southeast corner of New York experience the smallest change (0%–5%), while the area north of Maine experiences the largest change (20%–30%). Results for a return period of 100 years (Figure 8c) share quite similar spatial

patterns with an average increase of 11%. Interestingly, several stations around New Jersey and the southeast corner of New York even show some negative values (−5%–0%), indicating a decrease of PF estimates. This decrease of PF estimates in some areas is also visible for the LOCA results for the return periods of 25 and 100 years (Figure 8e,g). In general, the results based on the LOCA data show relatively smaller increases with an average increase of 8% for a return period of 25 years and 10% for a return period of 100 years. The spatial distribution is also more scattered than for the UWPD data. PF estimates increase more dramatically by the second half of the 21st century. The UWPD data show an increase of 15% (Figure 8b) and 17% (Figure 8d) for return periods of 25 years and 100 years, respectively, while the LOCA data show an increase of 12% (Figure 8f) and 13% (Figure 8h).

Under the high emission scenario (RCP8.5), the magnitude of PF estimate changes is in general larger than for the low emission scenario. For the first half of the 21st century, the results based on the UWPD data show that PF estimates increase by a mean value of 12% for a return period of 25 years (Figure 9a) and 14% for a return period of 100 years (Figure 9c). Again, the LOCA data show smaller increases of 8% and 9% for return periods of 25 years and 100 years, respectively (Figure 9e,g). By the second half of the 21st century, however, the magnitudes increased quite dramatically, with both the UWPD and LOCA data sets showing average increases of over 20% (Figure 9b,d,f,h). For some stations around northern Maine, the UWPD data even show an increase of 50% for a return period of 100 years (Figure 9d). The LOCA data also have several stations with increases of over 50% (Figure 9h), but these stations are not as concentrated in one place as they are for the UWPD data. These location differences may be due to the ability of the models or downscaling schemes to simulate extratropical cyclones, which have been shown to be the main cause of extreme precipitation events in the Northeastern US [42]. The number of these cyclones is indeed projected to increase in the future [43]. Nevertheless, the PF estimate increases seem to be robust even though the distributions are not spatially uniform nor the same across the different downscaling data sets. Another way to look into the PF estimate changes is through the so-called exceedance probability, which will be discussed in the next section.

## 5. Changes to Exceedance Probabilities

The exceedance probability is the probability that a given rainfall total accumulated over a given duration (daily in our paper) will be exceeded in any one year. Compared with the related concept of a return period, it may sometimes be better to use the exceedance probability instead since many misconceptions are often associated with return periods. For example, there may be a misunderstanding that once an event happens, a certain number of years (the return period) must pass for an event of that magnitude to happen again. However, extreme events can happen in any year, regardless of how much time has elapsed since the previous one, an idea more easily conceptualized by the exceedance probability. These misconceptions associated with return periods are already discussed in depth [44]. Nonetheless, the term "return period" is still widely used in the literature, which is one reason why we employed it in our analyses in the previous sections. To complement these analyses, in this section, we examined future extreme precipitation events through the concept of exceedance probability. Specifically, we examined what the exceedance probabilities of the historical PF estimates will be in the future.

The probability density function for a GEV distribution associated with an AMS is given by:

$$P_{pdf} = \frac{1}{\sigma} t(x)^{1+\xi} e^{-t(x)}, \tag{2}$$

where $t(x) = \left(1 + \xi\left(\frac{x-\mu}{\sigma}\right)\right)^{-\frac{1}{\xi}}$, if $\xi \neq 0$ and $t(x) = e^{-\frac{(x-\mu)}{\sigma}}$ if $\xi = 0$. $\mu \in \mathbb{R}$ is the location parameter; $\sigma > 0$ is the scale parameter; and $\xi \in \mathbb{R}$ is the shape parameter.

Note that sometimes the shape parameter, $\xi$, is defined differently, often having opposite signs in different sources [40,45,46]. The cumulative density function (CDF) can be written as:

$$P_{cdf} = e^{-t(x)}. \tag{3}$$

The relevant exceedance probability, $p$, for a certain value, $x$, that is, the probability that a given year's maximum daily precipitation is larger than $x$, is given by:

$$p = 1 - P_{cdf} = 1 - e^{-t(x)}. \tag{4}$$

Since the CDF is invertible, the quantile function for the GEV distribution has an explicit expression. For any given exceedance probability, $p$, the related quantile is given by:

$$x = P_{cdf}^{-1}(1 - p). \tag{5}$$

For a given return period, $T$, in years, the exceedance probability for the PF associated with that period is $\frac{1}{T}$ (remembering that only one value per year is used in an AMS). Therefore, the PF can be calculated by plugging $\frac{1}{T}$ into Equation (5):

$$PF = P_{cdf}^{-1}\left(1 - \frac{1}{T}\right). \tag{6}$$

For any given AMS data, the GEV distribution parameters, $\mu$, $\sigma$, and $\xi$, can be estimated by using methods, such as the maximum likelihood method and the L-moment method [36,42]. Parameters estimated based on the historical data are labeled here ($\mu_{hist}$, $\sigma_{hist}$, $\xi_{hist}$, $PF_{hist}$) and parameters estimated based on the projected data are labeled ($\mu_{future}$, $\sigma_{future}$, $\xi_{future}$). Likewise, PF estimates calculated based on the historical and projected data are labeled $PF_{hist}$ and $PF_{future}$, respectively. To examine how the exceedance probabilities of the historical PF estimates will change in the future, we can compare the following:

$$P_{hist} = 1 - P_{cdf}\left(\mu_{hist}, \sigma_{hist}, \xi_{hist}, PF_{hist}\right) = \frac{1}{T}, \tag{7}$$

$$P_{future} = 1 - P_{cdf}\left(\mu_{future}, \sigma_{future}, \xi_{future}, PF_{hist}\right). \tag{8}$$

In Equation (7), we calculate the historical exceedance probability, $P_{hist}$, using distribution parameters estimated from the historical data and the $PF_{hist}$ corresponding to a given time period $T$. In Equation (8), we instead calculate the future exceedance probability, $P_{future}$, using distribution parameters estimated from the projected future data, while still using the historical $PF_{hist}$. Since $\mu$, $\sigma$, and $\xi$ are different in each equation, we can expect different exceedance probabilities from each for the same PF estimates (or the same return periods).

Figure 10 shows averaged exceedance probabilities for the Northeastern US calculated from Equations (7) and (8) using the UWPD and LOCA data under the RCP4.5 and RCP8.5 emission scenarios. As expected, the projected exceedance probabilities (black and green lines) are all larger than the corresponding historical exceedance probabilities (red lines). For example, the historical 2-year event (total daily rainfall ≈60 mm) has an exceedance probability of 0.5 for one year. Under the RCP4.5 scenario, the UWPD data (Figure 10a) show that this amount will have an exceedance probability of ≈0.59 for the first half of the 21st century and 0.65 for the second half of the 21st century. These changes are more dramatic for larger events. For a historical 25-year event (total daily rainfall ≈107 mm), the exceedance probability changes from ≈0.04 to ≈0.07 for the first half of the 21st century and to 0.09 for the second half. For a historical 100-year event (total daily rainfall ≈138 mm), the exceedance probability changes from ≈0.01 to ≈0.03 for the first half of the 21st century and to ≈0.04 for the second half. The future exceedance probabilities are almost quadruple their historical values. This means a 100-year event will become a 35-year (1/0.0286 precisely, Table 2) event for the first half of the 21st century and a 27-year (1/0.037 precisely) event for the second half of the 21st century. Under the RCP8.5 scenario, the UWPD data (Figure 10b) show that the exceedance probability for a 100-year event may increase to 0.05 for the second half of 21st century, which is five times larger than its historical value; a 100-year event will become a 19-year event. The LOCA data (Figure 10c,d) in general tell a

similar story, although the magnitudes are relatively smaller, probably related to the dry bias seen earlier in the data. Under the RCP8.5 scenario, the exceedance probability for the 100-year event (0.01) will increase to ≈0.04 for the second half of the 21st century, which is still quite a significant change. Different ensemble members indeed show different magnitudes (reflected by the short vertical lines in Figure 10 representing the ensemble spread), but overall, the signs of the changes are quite consistent, suggesting that the extreme events will become more frequent in the future.

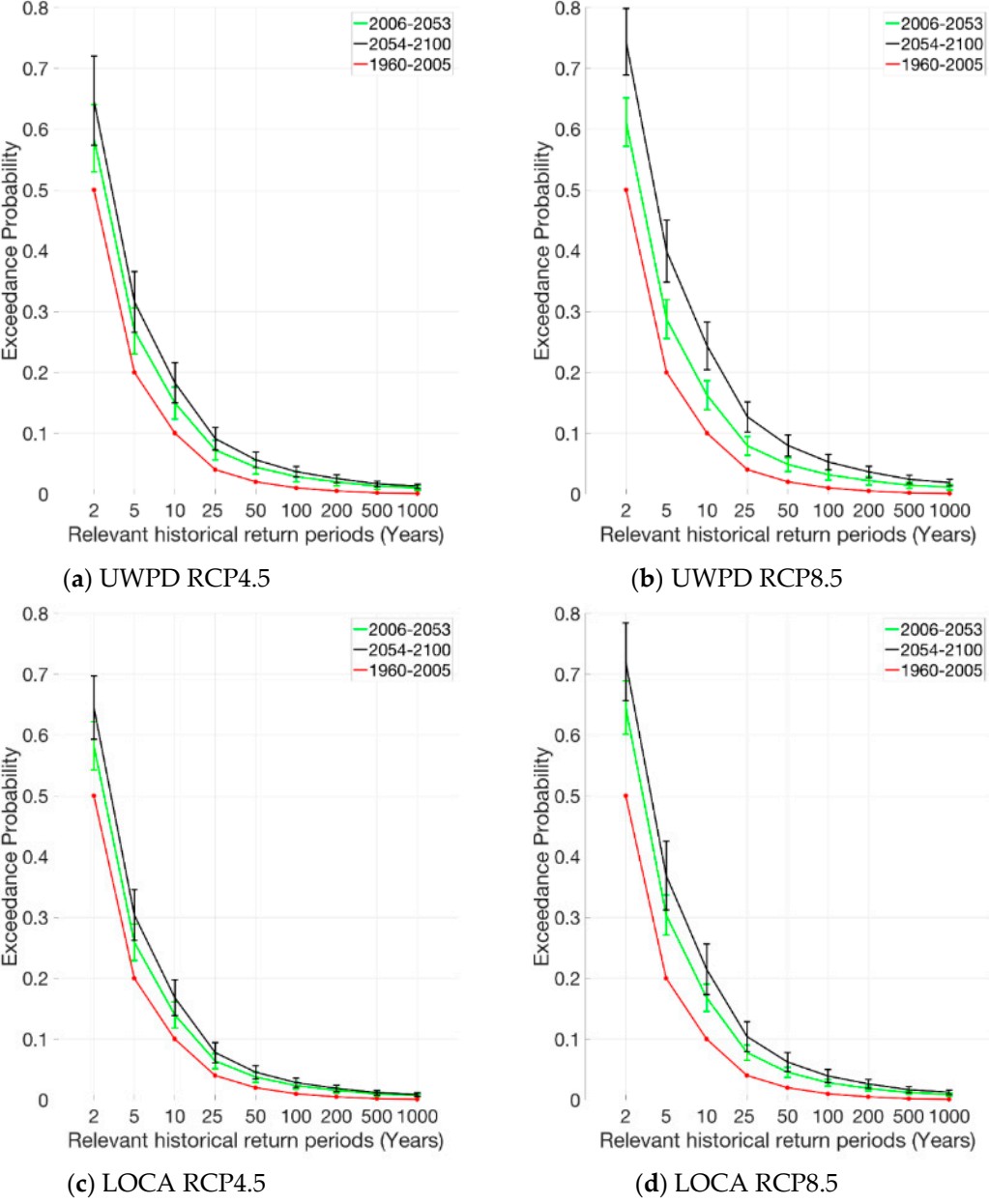

**Figure 10.** Northeastern US domain averaged exceedance probabilities (Equations (7) and (8)) for different relevant historical return periods (related to $PF_{hist}$) under different scenarios. Results for UWPD (upper; (**a**): RCP4.5; (**b**): RCP8.5) and LOCA (lower; (**c**): RCP4.5; (**d**): RCP8.5). The short vertical lines represent the uncertainties estimated by the ensemble spread. Red represents the results for the historical period (1960–2005), green for the first half of the 21st century (2006–2053), and black for the second half (2054–2100).

**Table 2.** Northeastern US domain averaged return periods with their uncertainties expressed as +/− one standard deviation of the ensemble under RCP4.5 and RCP8.5 scenarios.

| Relevant Historical Return Periods (Years) | | | 2 | 5 | 10 | 25 | 50 | 100 | 200 | 500 | 1000 |
|---|---|---|---|---|---|---|---|---|---|---|---|
| Projected return periods (UWPD) | RCP4.5 | 2006–2053 | 1.7 [1.6–1.9] | 3.7 [3.3–4.3] | 6.7 [5.7–8.1] | 13.8 [11.3–17.8] | 22.7 [18.0–30.6] | 34.9 [27.0–49.5] | 50.9 [38.5–75.3] | 77.2 [56.9–120.1] | 100.4 [72.9–161.3] |
| | | 2054–2100 | 1.5 [1.4–1.7] | 3.2 [2.7–3.8] | 5.5 [4.6–6.7] | 11.0 [9.1–13.9] | 17.8 [14.5–22.9] | 27.2 [22.0–36.0] | 39.5 [31.5–53.2] | 59.9 [47.0–82.5] | 77.8 [60.6–108.8] |
| | RCP8.5 | 2006–2053 | 1.6 [1.5–1.7] | 3.5 [3.1–3.9] | 6.2 [5.3–7.2] | 12.6 [10.5–15.7] | 20.6 [16.6–27.0] | 31.6 [24.8–43.6] | 45.9 [34.9–66.9] | 69.4 [51.0–108.3] | 90.0 [64.7–147.7] |
| | | 2054–2100 | 1.3 [1.2–1.5] | 2.5 [2.2–2.9] | 4.1 [3.5–4.9] | 7.9 [6.6–9.8] | 12.5 [10.3–16.0] | 19.0 [15.3–25.0] | 27.4 [21.8–37.0] | 41.4 [32.3–57.8] | 53.9 [41.6–76.5] |
| Projected return periods (LOCA) | RCP4.5 | 2006–2053 | 1.7 [1.6–1.8] | 3.9 [3.5–4.4] | 7.1 [6.2–8.4] | 15.6 [13.1–19.4] | 26.7 [21.7–34.5] | 42.7 [34.1–57.1] | 64.5 [50.7–88.7] | 102.0 [78.8–144.7] | 136.2 [104.0–197.1] |
| | | 2054–2100 | 1.5 [1.4–1.7] | 3.3 [2.9–3.8] | 5.9 [5.1–7.2] | 12.9 [10.6–16.4] | 22.0 [17.7–28.9] | 35.2 [27.9–47.9] | 53.3 [41.4–74.8] | 84.1 [63.6–123.9] | 112.0 [83.2–171.1] |
| | RCP8.5 | 2006–2053 | 1.7 [1.6–1.8] | 3.7 [3.3–4.2] | 6.9 [5.9–8.1] | 15.1 [12.7–18.7] | 26.0 [21.5–33.0] | 42.0 [34.1–54.6] | 63.7 [51.0–84.7] | 101.1 [79.8–138.0] | 135.2 [105.6–187.9] |
| | | 2054–2100 | 1.4 [1.3–1.5] | 2.7 [2.4–3.2] | 4.7 [3.9–5.8] | 9.6 [7.8–12.5] | 16.1 [12.8–21.5] | 25.4 [20.1–34.7] | 38.2 [29.9–52.9] | 60.0 [46.6–84.3] | 79.8 [61.7–113.1] |

## 6. Discussion and Conclusions

The observed global mean surface temperature has increased by about 1.0 °C (0.8–1.2 °C) above pre-industrial levels [47]. Associated with this temperature rise, the atmosphere's ability to hold water vapor has also increased, as described by the Clausius–Clapeyron relationship, implying a possible increase in extreme precipitation events [48]. Indeed, observational data show that extreme precipitation events in many parts of the world have increased in intensity dramatically over the past several decades, raising concerns about future changes. To address these concerns, many approaches based on observational and climate model data have been used. Despite significant uncertainties [49], climate models are used in numerous research publications to study projected climate changes and their consequences [2].

In contrast to the traditional approach of studying per-event precipitation intensities [50], in this study, based on observational and model projected data, we examined the point precipitation frequency (PF) estimates for the periods of 1960–2005, 2006–2053, and 2054–2100 in the Northeastern US, where the magnitudes of extreme precipitation events are projected to increase the most dramatically in the US [5]. Spatially, the Northeastern US is characterized by a highly diverse climate influenced by several geographic factors, such as the Atlantic Ocean to the east, the Great Lakes to the west, and the Appalachian Mountains to the south. The mountain ranges often block air flow, leading to local enhancement of precipitation through orographic lift [4]. To study the extreme precipitation events in such a complex region, we evaluated three commonly used downscaled data sets (LOCA, BCCAv2, and NA-CORDEX), along with a probabilistic downscaled data set developed specifically for this study (UWPD), on their ability to reproduce observed PF estimate features. This analysis was performed using four approaches, including one AMS-based and three PDS-based methods, but the variability in the projected PF estimates among these methods was not deemed significant. The data set with the highest historical accuracy statistics (UWPD) and one of the most commonly used data sets (LOCA) [2] were selected to analyze their projected PF estimates. Our results demonstrate that these PF estimates are likely to continue increasing, regardless of the emission scenario. The largest increases were in the northernmost part of the domain while the southern areas showed smaller increases. Under the RCP4.5 scenario, for the UWPD data set, the average projected increases to the 100-year event are 10.79% for the first half of the 21st century and 16.90% for the second half. The corresponding increases under RCP8.5 are 14.18% for the first half of the 21st century and 26.75% for the second half. This is generally consistent with the results of previous CMIP3 analyses of annual precipitation increases [4]. Our results also highlight how the associated exceedance probabilities of the historical PF estimates will change with projected climates. Notably, an event with a current exceedance probability of 0.01 (a 100-year event) may have an exceedance probability for the second half of the 21st century of ≈0.04 (a

27-year event) under the RCP4.5 scenario and ≈0.05 (a 19-year event) under RCP8.5, according to the UWPD data. It is thus vital to understand how these extreme precipitation events are projected to change so that the design decisions of today can serve to protect the socio-economic and environmental interests of the future.

**Author Contributions:** S.W. and M.M. designed the research framework. S.W. carried out the analysis. M.M. acted as the project leader. D.L. generated the UWPD downscaled data set and provided the description of this data. J.R.A. and K.G. participated in discussions and manuscript preparation.

**Funding:** The paper is supported by NOAA/UCAR Contract # SUBAWD000255.

**Acknowledgments:** We are grateful to Sanja Perica for her valuable comments and suggestions. We also thank Michael St Laurent for sharing the NOAA Atlas 14 observational data. Linda Mearns provided valuable discussions on the NA-CORDEX data; Katja Winger helped with storage issues on the NA-CORDEX data. David W. Pierce and Ben Livneh provided explanations about the LOCA data set, and Kenneth Nowak explained the BCCAv2 data set. Discussions with Brian Beucler during our online meeting were also quite helpful. Lisa Sheppard (Illinois State Water Survey) provided editorial assistance.

**Conflicts of Interest:** The authors declare no conflict of interest.

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
