# Peer review of "A Comparative Analysis of the Historical Accuracy of the Point Precipitation Frequency Estimates of Four Data Sets and Their Projections for the Northeastern United States"

_water, doi:10.3390/w11061279_

Round 1
Reviewer 1 Report
The topic well fits the journal and it is certainly of interest for many environmental fields, from climatic to hydrological applications. This manuscript describes an effort to evaluate the extreme precipitation events in Northeastern United States through comparison against long term station’s records. The manuscript contains a lot of details and well describes. In overall, the research strategy is well designed and carried out successfully. Although, I find a lot of merit to this study, I do have some comments and questions. Therefore, the reviewer recommends that this manuscript be published after some minor language correction and clarification specified below:
General comments:
While I find a lot of merit to this study, I do have some comments and questions listed below:
The authors used the downscaled products and compared them with ground observations. But they didn’t mention that these downscaled products are not yet adequate for ground-based comparison. Because the spatial resolution of the products they used are still too coarse, particularly for small-scale basin studies. Precipitation might occur in a smaller scale than the model products grid points. The model represents the areal average precipitation, while the ground precipitation indicates point precipitation measurement. However, you can make an average of those stations that are fallen within a model’s pixel and then do the comparison.
Moreover, in this study the authors selected the thresholds based on the precipitation frequency. It would be valuable to add discussion about the intensity of precipitation per-event too, or combination of them. As an example please have a look here https://doi.org/10.1016/j.atmosres.2018.02.020.
Specifics comments:
Line 62-64: Please explain more, how the downscaling method can add uncertainty to the models? Since the aim of the downscaling is to give more details in fine-scale and subsequently increase the accuracy of the models.
Line 66: “… throughout the Northeast” of where?
Line 68: “… including three commonly used ones, and a fourth probabilistic data set developed for this study”. Please name them.
Line 90: at the end you used 758 stations or used all stations’ data (1218)?
Line 102-104: I suggest to give a reference to Dr. David Lorenz and say for example “… was generated by Lorenz et. al. [reference]”.
Line 114-116: Could you explain why they used logistic and generalized gamma distribution?
Line 122-123: Which methods did you used to correct the time of observation? Do these stations’ data have been already quality controlled? Was it a large number of stations removed due to large biases?
Line 133: Which interpolation techniques did you used and why?
Line 290: In figure 3, I suggest to put the OP, UWPD and LOCA as the title of the “y” axis, with vertical position.
Figures 8 and 9: Could you please explain what would be the reason, why the relative PF difference ratio in these two figures are less (close to zero) over the south and southwest and are increase over the northeast part of your domain? Please revised this figure, the title of “y” axis overlapped the y labels, etc.
The topography map of the region might be useful for the readers.
Author Response
We thank the reviewers for their valuable comments and suggestions (in red). Here are our point-by-point replies (in blue). We also notice that the layout and line numbers of Word file may change during the online submission process. The line numbers mentioned in this reply may not match exactly that the reviewer receives. Therefore, we uploaded a Word file with all changes tracked.
Reply to Review 1.
The topic well fits the journal and it is certainly of interest for many environmental fields, from climatic to hydrological applications. This manuscript describes an effort to evaluate the extreme precipitation events in Northeastern United States through comparison against long term station’s records. The manuscript contains a lot of details and well describes. In overall, the research strategy is well designed and carried out successfully. Although, I find a lot of merit to this study, I do have some comments and questions. Therefore, the reviewer recommends that this manuscript be published after some minor language correction and clarification specified below:
General comments:
While I find a lot of merit to this study, I do have some comments and questions listed below:
The authors used the downscaled products and compared them with ground observations. But they didn’t mention that these downscaled products are not yet adequate for ground-based comparison. Because the spatial resolution of the products they used are still too coarse, particularly for small-scale basin studies. Precipitation might occur in a smaller scale than the model products grid points. The model represents the areal average precipitation, while the ground precipitation indicates point precipitation measurement. However, you can make an average of those stations that are fallen within a model’s pixel and then do the comparison.
Moreover, in this study the authors selected the thresholds based on the precipitation frequency. It would be valuable to add discussion about the intensity of precipitation per-event too, or combination of them. As an example please have a look here https://doi.org/10.1016/j.atmosres.2018.02.020.
The reviewer raises a very important question whether or not the downscaled products represent a cell/pixel averaged value or a point value. Global climate model (GCM) outputs represent pixel averaged values. The regional model output used in this study, the dynamically downscaled data set NA-CORDEX, also represents pixel averaged values. However, this may not be true for statistically downscaled data. We contacted the research teams who developed the statistically downscaled data sets, and here are the answers we got from them:
“In the most basic sense, the interpolated data represent point values that are located on the center of the regular grid coordinates. They are often used as grid means, despite this.
”--- Ben Livneh: Ben.Livneh@colorado.edu on LOCA data
“Each cell has a corresponding value that represents an average condition for that area with respect to a given variable/parameter (e.g. temperature).
”--- Kenneth Nowak knowak@usbr.gov on BCCA data
Therefore, the statistically downscaled data can represent either pixel averaged values or single point values. It should be noted that the resolutions of the downscaled products we used in the paper are 1/10 degrees (UWPD), 1/16 degrees (LOCA), 1/8 degrees (BCCAv2) and 0.22 degrees (or 0.44 degrees for one model) (NA-CORDEX). For the UWPD and LOCA data sets, which already represent point values, no correction was necessary. For the BCCAv2 and NA-CORDEX data, which represent pixel means, we corrected the results by applying empirical area correction factors according to Miller et al. 1973 (Line 258-262). Adding more analysis on the intensity of precipitation per-event would be very interesting, but its focus would be different than that of the research presented herein. Nevertheless, we mentioned this reference (Line 726) to further clarify the scope of our research.
Specifics comments:
Line 62-64: Please explain more, how the downscaling method can add uncertainty to the models? Since the aim of the downscaling is to give more details in fine-scale and subsequently increase the accuracy of the models.
Downscaling methods are often categorized into two major groups: statistical and dynamical. Both statistical and dynamical downscaling can be performed using many different approaches, produce different results, and generally have different accuracies. This is the main source of additional uncertainty introduced by downscaling. However, downscaling is necessary for the study of hydrologic applications, the main focus of this paper. Large grid-cells cannot be used in most hydrological studies, particularly those dealing with small-scale events. Downscaling increases the data resolution so that the results can be usable in small-scale studies.
Line 66: “… throughout the Northeast” of where?
Changed to “Northeastern US” in Line 76 now.
Line 68: “… including three commonly used ones, and a fourth probabilistic data set developed for this study”. Please name them.
We added a reference to Table 1 in line 79 now. The data sets are those listed in the table.
Line 90: at the end you used 758 stations or used all stations’ data (1218)?
We used 753 stations as stated in the paper (Lines 120-121). Figure 1 has also been modified to show only the locations of analyzed stations.
Line 102-104: I suggest to give a reference to Dr. David Lorenz and say for example “… was generated by Lorenz et. al. [reference]”.
Added in lines 141-142 now.
Line 114-116: Could you explain why they used logistic and generalized gamma distribution?
Logistic regression was used because it is the appropriate regression analysis for binary data. Several potential link functions were explored for logistic regression including the probit and several skewed link functions. It was found that the standard logit link function fit the observed data best. For the precipitation amount, the exponential, gamma, Weibull and generalized exponential distributions were insufficiently flexible to capture the precipitation extremes. Therefore, the generalized gamma distribution, which has two shape parameters instead of one or zero, was used. The generalized gamma distribution includes the gamma and the Weibull distribution as special cases. Note, a single generalized gamma distribution is not used for all days, instead the parameters of the generalized gamma distribution depend on the daily varying large-scale atmospheric conditions. We have added this information to lines 156-165.
Line 122-123: Which methods did you used to correct the time of observation? Do these stations’ data have been already quality controlled? Was it a large number of stations removed due to large biases?
The methods to correct the time of observation are complex and beyond the scope of this work. A brief description follows. First the subset of stations with hourly data was analyzed to find an empirical method for determining the DIFFERENCE in hour of observation between daily precipitation stations. This is done by "artificially" aggregating the hourly data into daily accumulations for all possible combinations of the hour of observation. It was found that a method involving time lagged correlations between daily precipitation resulted in a good scheme for determining the difference in hour of observation. Next, time lagged correlations were calculated between all pairs of daily stations within a 70 km radius. The scheme developed on the hourly data was then used to estimate the hour difference between all station pairs. Next, this calculated difference was compared to that reported in the COOP metadata. The reported difference for some stations agrees very well with that estimated by our method. For other stations, however, the difference is substantial. For example, some stations are off by 24 hours because the COOP volunteers apparently did not understand that the reported date should be that of the observation date even if the majority of the 24 hour period is in the previous day. In addition, some observers do not report when they switch from evening to morning observations and vice versa. So, for our methodology we rank the stations by the degree of disagreement between the reported and estimated hour difference. The station with the worst agreement is then adjusted to optimally come into best agreement with our estimated hour difference. After the adjustment, the degree of disagreement between all station pairs that involve this station are also adjusted to take into account the revised estimate. Next the stations are ranked again by the degree of disagreement and the worst station is adjusted. The process is continued until the disagreement between all station pairs is less than or equal to 3 hours. This method is repeated separately for each month in the record because the hour of observation can change multiple times at any point in the record. Due to the month by month nature of the above adjustment, a scheme was devised to identify the "change points" in time when hour of observation likely changed for each pair of stations. Then instead of calculating the lagged correlations individually for each month, the lagged correlations are calculated over the entire time period between "change points". Typically the identified change points are spaced at least a decade apart, which enables a robust calculation of the time lagged correlation. 21% of stations were removed due to the biases described in the following reference. This has been added in line 171.
C. Daly; W. P. Gibson; G. H. Taylor; M. K. Doggett; and J. I. Smith. Observer bias in daily precipitation measurements at United States cooperative network stations. 2007, Bull. Amer. Meteor. Soc., 88, pp899–912.
Line 133: Which interpolation techniques did you used and why?
We refer the reviewer to our following cited paper for an explanation of our interpolation techniques and our reasons for using them: M. Notaro; D. J. Lorenz; C. Hoving; and M. Schummer. Twenty-first-century projections of snowfall and winter severity across central-eastern North America. 2014, J. Clim., 27, 6526–6550.
Line 290: In figure 3, I suggest to put the OP, UWPD and LOCA as the title of the “y” axis, with vertical position.
Revised according to your suggestions. Thanks.
Figures 8 and 9: Could you please explain what would be the reason, why the relative PF difference ratio in these two figures are less (close to zero) over the south and southwest and are increase over the northeast part of your domain? Please revised this figure, the title of “y” axis overlapped the y labels, etc.
Spatially, the Northeastern US is characterized by a highly diverse climate influenced by several geographic factors such as the Atlantic Ocean to the east, the Great Lakes to the west and the Appalachian Mountains to the south. The mountain ranges often block air flow leading to local enhancement of precipitation through orographic lift (Kunkel et al. 2013). Weather systems, including extratropical cyclones associated with fronts (FRT) or centers of low pressure (ETC) as well as tropical cyclones (TC), have been shown to be the main cause of extreme precipitation events in this region (Kunkel et al 2012). To understand the mechanisms behind the spatial distributions of the PF increases, including this apparent southwest to northeast gradient, we would need rigorous analyses of all these system types in terms of their intensity, numbers and locations under both RCP4.5 and RCP8.5. This is very important and deserves more attention, perhaps in another paper.
As the reviewer suggested, we have revised the figure.
K. E. Kunkel; L.E. Stevens; S.E. Stevens; L.Q. Sun; E. Janssen; D. Wuebbles; J. Rennells; A. DeGaetano; and J.G. Dobson. Regional climate trends and scenarios for the U.S. 2013, National Climate Assessment: Part 1—Climate of the Northeast U.S., NOAA Tech. Rep. NESDIS 142-1, 80 pp.
K. E. Kunkel; D. R. Easterling; D. A. R. Kristovich; B. Gleason; L. Stoecker; and R. Smith. Meteorological causes of the secular variations in observed extreme precipitation events for the conterminous United States. 2012, J. Hydrometeor., 13, 1131–1141.
The topography map of the region might be useful for the readers.
A clearer topography map is now shown in Figure 1. Thanks.

Reviewer 2 Report
Manuscript ID water-508354, entitled “A comparative analysis of the hindcast accuracy of the point precipitation frequency estimates of four data sets and their projections for the Northeastern United States” by Wu et al.
This manuscript presents a relevant study, based on a large amount of data and analysis which in my opinion deserves being published after the authors address the following caveats.
The title should be revised as the terms "hindcast accuracy" are not clear/defined nor further used through the manuscript. Additionally, further discussion should be presented on the "point precipitation" representation when using some very coarse models' outputs.
Abstract should be improved. Indication of methods used. Quantification of results should be included.
Ln 47 – it should be defined what “precipitation frequency (PF)” is
Ln 66 – for an international scientific community, “Northeast” should be replaced by “Northeastern United States” or similar expression
Ln 83 – section should present some conclusions.
Ln 89 – please confirm/explain why reference [21] is made to Volume 11: “(…) Volume 11 Version 2.0: Texas. 2018”
Ln 90 – Again, there is no information regarding the precipitation data frequency. Is this study based on hourly, daily, accumulated totals, daily maxima values, monthly data or yearly maximum? More detail should be provided on data used. Similar analysis with different results could be performed with either above mentioned data.
Ln 99 - Table 1: more information should be given on the acronyms (or a reference).
Ln 100-104 – It is not clear from this paragraph if this work has been previously published. If it has, a reference is missing. If not, all validation procedure and results should be presented or published previously to being used.
Ln 104-108 – a reference is missing
Ln 109-112 – These assumptions must be justified. How do results on this manuscript depend on these assumptions?
Ln 129-131 – A reference should be included – or results should be presented and discussed.
Ln 132-148 – References should be included though this paragraph.
149-151 – The manuscript should have all information without only references to websites. These change frequently and become outdated. Furthermore, in this case, many links are “Under construction” (e.g. Data and Data Quality Control)
In summary, information on UWPD (University of Wisconsin Probabilistic Downscaling) data set should be presented as in LOCA data (ln 171-176).
Ln 173 – “results from 32 models” – As previously stated, Table 1 should be clarified/explained. Are results used provided by “32 models”? Or by 32 simulations from # models? Similarly more information should be given to all ensemble members used on all 4 dataset groups (Ln 178) .
Ln 203-204 – This information should be given in the abstract and before in the manuscript. See comments on Lns 47 and 90.
Ln 209 – please give further information on “retaining the values over a preselected threshold”. How is the threshold obtained? Is there a sensitivity analysis on the results to the threshold chosen? Since this is explained later, a reference should be included here (e.g. [32])
Figure captions should be improved. On Figure 2 the meaning of the different colors used should be included. In the discussion of Figure 2 quantification of thresholds should be performed. Units should be presented in SI Units (mm/day).
Ln 269-270 – “For this analysis, the closest model grid point to a rain gage station is selected as its representation in the model”. Please correct to “rain gauge”; this option should be discussed and a reference included. Depending on orography and local characteristics this may not be the best option.
Ln 272 - climatological means
Ln 272 – do you mean “observed and modeled AMS and PDS” or “AMS and PDS of observed and modelled data”
Ln 282 – again, as previously stated, what means are you referring to? This should be clearly mentioned over the text “climatological daily means” or “climatological maximum daily means”
Ln 296 – similar comment as in Ln 272. As far as I understand you don’t model the PF. You estimate the PF for modelled data.
Ln 303-304 – “to get a complete picture.” Too colloquial. Please rephrase.
Figure captions should be improved. On Figure 4 the meaning of the different colors used should be included. Figure 5 caption could be replaced by “Similar to Figure 4 but for LOCA data”
Figure 4 and 5 should have the same scales so that the panels could be comparable: e.g. in 25-yr axes should be from 2 to 7.5 inches both in Figure 4e and 5e.
Ln 356-359 – again, results should be presented in SI Units and in percentages.
Ln 371 – equations for computing “correlation coefficients and RMSE” or at least a reference should be included in section 2.2 Methodologies
Ln 412-420 – 30-yr periods are commonly used, e.g. near future as 2006-2035 and end of the 21st century as 2070-2099.
Figures 8 and 9 color schemes are confusing. Cold colors should represent a decrease change and warm colors should represent an increase change (or vice-versa). With this colorbar it is visually difficult to distinguish between negative and positive values (e.g. -0.1 and 0.1)
Ln 473-478: There is much literature on how extreme extratropical cyclones may change under future scenarios (e.g. Ulbrich et al. 2013) so further references and information on this could be added to corroborate the discussion of these results. However, some discussion should also include the correlation between obtained results and topography.
Ln 490-491 – “Nonetheless, these errors are still widely used in the literature, which is one reason why we employ them in our analyses in the previous sections” Please rephrase: what do you mean by “them”? The “errors”?
Ln 513 – “etc.” should be avoided
Ln 548 – “model spread”, as previously stated, “model” and “ensemble” are not synonymous in this study. The text should be revised carefully.
Figure 10 caption: please indicate what colors represent.
Reference:
Ulbrich et al. (2013) Are Greenhouse Gas Signals of Northern Hemisphere winter extra-tropical cyclone activity dependent on the identification and tracking algorithm? Meteorologische Zeitschrift, Vol. 22, No. 1, 61–68 DOI 10.1127/0941-2948/2013/0420
Author Response
We thank the reviewers for their valuable comments and suggestions (in red). Here are our point-by-point replies (in blue). We also notice that the layout and line numbers of Word file may change during the online submission process. The line numbers mentioned in this reply may not match exactly that the reviewer receives. Therefore, we uploaded a Word file with all changes tracked.
Reply to Review 2.
Summary of replies.
This manuscript presents a relevant study, based on a large amount of data and analysis which in my opinion deserves being published after the authors address the following caveats.
The title should be revised as the terms "hindcast accuracy" are not clear/defined nor further used through the manuscript. Additionally, further discussion should be presented on the "point precipitation" representation when using some very coarse models' outputs.
By “hindcast accuracy,” we refer to our comparisons between the observations and downscaled model data sets for the historical period of 1960-2005. The data sets that match the observations the best have the highest hindcast accuracies. For clarity, we have changed “hindcast accuracy” to “historical accuracy” in the title, and throughout the text.
It should be noted that the resolutions of the downscaled data sets we used are not too coarse even though they were generated using the output of coarse models. Actually, the resolutions are similar to or sometimes even finer than high resolution satellite observations, which are commonly used for comparisons with ground-based observations. Nevertheless, we indeed added more analysis to address the possible representation errors for the two relatively coarse resolution data sets, namely BCCAv2 (1/8 degrees) and NA-CORDEX (0.22 and 0.44 degrees), by adding area correction factors as discussed in Lines 259-263.
Abstract should be improved. Indication of methods used. Quantification of results should be included.
We have revised the abstract by adding descriptions of our methods and results in lines 15-17 and 26-29, respectively.
Ln 47 – it should be defined what “precipitation frequency (PF)” is
Added. Lines 57-58.
Ln 66 – for an international scientific community, “Northeast” should be replaced by “Northeastern United States” or similar expression
Changed to “Northeastern US” in Line 76 now.
Ln 83 – section should present some conclusions.
Added. Thanks. Line 113
Ln 89 – please confirm/explain why reference [21] is made to Volume 11: “(…) Volume 11 Version 2.0: Texas. 2018”
Corrected. Thanks. We have the right reference now.
[23] S. Perica, S. Pavlovic; M. S. Laurent; C. Trypaluk; D. Unruh; D. Martin; O. Wilhite, NOAA Atlas 14 Precipitation-Frequency Atlas of the United States Volume 10 Version 3.0: Northeastern States. 2019.
Ln 90 – Again, there is no information regarding the precipitation data frequency. Is this study based on hourly, daily, accumulated totals, daily maxima values, monthly data or yearly maximum? More detail should be provided on data used. Similar analysis with different results could be performed with either above mentioned data.
We used daily precipitation data. This has been clarified in lines 15, 269, 271 and 574.
Ln 99 - Table 1: more information should be given on the acronyms (or a reference).
We have added another reference. Lines 221-222.
Ln 100-104 – It is not clear from this paragraph if this work has been previously published. If it has, a reference is missing. If not, all validation procedure and results should be presented or published previously to being used.
We have added the reference. Line 142
Ln 104-108 – a reference is missing
We have added the reference. Line 144
Ln 109-112 – These assumptions must be justified. How do results on this manuscript depend on these assumptions?
The assumption of a Gaussian distribution may be the most widely used one. Our comparative studies between the UWPD data and the observations for the historical periods can actually be treated as a test of this assumption. The good performance of the UWPD data in terms of reproducing the observed features of the PFs indicate that this assumption is likely valid. Using another assumption may result in a different outcome. Unfortunately, we could not carry out systematic tests of other possible assumptions due to time constraints.
Ln 129-131 – A reference should be included – or results should be presented and discussed.
We have added the reference to line 179 now.
Ln 132-148 – References should be included though this paragraph.
We have added references [14, 17-18, 28].
149-151 – The manuscript should have all information without only references to websites. These change frequently and become outdated. Furthermore, in this case, many links are “Under construction” (e.g. Data and Data Quality Control)
We have added more references and technique details, lines 135-143, 156-165, 171, 216. The website serves as a complementary resource since it has extra information not included here such as a more detailed introduction to the data set and the research works that have utilized and evaluated the UWPD data in many different areas. Because the UPWD data, particularly the CMIP5 version, is relatively new, many links on the website are still under construction. This manuscript itself, through its comparisons with the observations, also serves as an evaluation of this data set.
In summary, information on UWPD (University of Wisconsin Probabilistic Downscaling) data set should be presented as in LOCA data (ln 171-176).
As the reviewer suggested, we have added more details about the UWPD data. We tried to provide as much information about the UWPD data as we could in the space we had, since it is a relatively new data set without as much documentation available as the other sets. In contrast, the LOCA data have been around longer and have much more extensive documentation available. Therefore, the LOCA data may not need as much introduction as the UWPD data.
Ln 173 – “results from 32 models” – As previously stated, Table 1 should be clarified/explained. Are results used provided by “32 models”? Or by 32 simulations from # models? Similarly more information should be given to all ensemble members used on all 4 dataset groups (Ln 178) .
We used one simulation of each of the 32 models. This has been clarified in lines 227-228, 231.
Ln 203-204 – This information should be given in the abstract and before in the manuscript. See comments on Lns 47 and 90.
Added. Thanks. Lines 16-17
Ln 209 – please give further information on “retaining the values over a preselected threshold”. How is the threshold obtained? Is there a sensitivity analysis on the results to the threshold chosen? Since this is explained later, a reference should be included here (e.g. [32])
Added. Line 276. Thanks.
Figure captions should be improved. On Figure 2 the meaning of the different colors used should be included. In the discussion of Figure 2 quantification of thresholds should be performed. Units should be presented in SI Units (mm/day).
Figure captions have been improved. All of our results and figures have been changed to SI Units.
Ln 269-270 – “For this analysis, the closest model grid point to a rain gage station is selected as its representation in the model”. Please correct to “rain gauge”; this option should be discussed and a reference included. Depending on orography and local characteristics this may not be the best option.
All instances of “gage” have been corrected to “gauge”.
Two of our selected data sets, namely UWPD and LOCA, have higher spatial resolutions than the other two data sets (BCCAv2 and NA-CORDEX). The possible maximum distances between the selected model grid points and rain gauge stations are
0.707*6356 km (Earth’s radius)*cos(40o)*(0.1/180*pi) = 7.18 km for the UWPD data
0.707*6356 km (Earth’s radius)*cos(40o)*(1/16/180*pi) = 4.50 km for the LOCA data
The possible minimum distances between the selected model grid points and rain gauge stations are
0.707*6356 km (Earth’s radius)*cos(45.5o)*(0.1/180*pi) = 4.88 km for the UWPD data
0.707*6356 km (Earth’s radius)*cos(45.5o)*(1/16/180*pi) = 2.81 km for the LOCA data
The mean distances are 5.28 km for the UWPD data and 3.13 km for the LOCA data. While this is a common problem in similar analyses such as those based on high resolution satellite data, we also recognize that selecting the closest grid point, particularly in mountainous areas, may not be the best option because it adds uncertainty to the results. A way to alleviate this issue could be the use of a weighted average, but this tends to smooth out the extreme events of interest. Given that the purpose of this study is to analyze average changes based on 753 stations, we assumed that the uncertainty caused by our gauge and grid point matching scheme would not affect the main conclusions of this study.
Ln 272 - climatological means
We have changed all instances of “climatology means” to “climatology”.
Ln 272 – do you mean “observed and modeled AMS and PDS” or “AMS and PDS of observed and modelled data”
Corrected in line 363 now. We meant the latter.
Ln 282 – again, as previously stated, what means are you referring to? This should be clearly mentioned over the text “climatological daily means” or “climatological maximum daily means”
We changed “climatology mean values” to “climatology”. As we have clarified, we used daily mean values.
Ln 296 – similar comment as in Ln 272. As far as I understand you don’t model the PF. You estimate the PF for modelled data.
Changed to “Comparison of PF estimates of the observed and modeled data” now in line 393. Thanks.
Ln 303-304 – “to get a complete picture.” Too colloquial. Please rephrase.
Changed to “to get a more complete understanding” in line 401 now.
Figure captions should be improved. On Figure 4 the meaning of the different colors used should be included. Figure 5 caption could be replaced by “Similar to Figure 4 but for LOCA data”
Figure 4 and 5 should have the same scales so that the panels could be comparable: e.g. in 25-yr axes should be from 2 to 7.5 inches both in Figure 4e and 5e.
All figure captions have been improved as suggested. Scales have been fixed. All the units have been changed to SI units (mm).
Ln 356-359 – again, results should be presented in SI Units and in percentages.
Corrected.
Ln 371 – equations for computing “correlation coefficients and RMSE” or at least a reference should be included in section 2.2 Methodologies
A reference was added to line 470.
Ln 412-420 – 30-yr periods are commonly used, e.g. near future as 2006-2035 and end of the 21st century as 2070-2099.
The selection of data set lengths of 30-yr, 48-yr (in our paper) or other lengths (e.g. 20-40 yr in [14]) is arbitrary. For shorter lengths, the stationarity assumption is more appropriate, but the frequency estimates for extreme events (e.g. a 100-year event) based on those short records is less reliable. On the other hand, longer data sets provide more reliable frequency estimates, but the assumption of stationarity is less appropriate. Since some of our target return periods were 50 years or longer, we felt that utilizing more data was necessary since that typically leads to more significant results for these longer periods. Based on our familiarity with the data sets used in this study, we tend to believe that using 30-yr periods instead would not produce dramatically different trends in the projected changes to heavy precipitation.
[14] M. Markus; J. Angel; G. Byard; S. McConkey; M.ASCE4; C. Zhang ; X.M. Cai; M.ASCE ; M. Notaro ; and M. Ashfaq. Communicating the Impacts of Projected Climate Change on Heavy Rainfall Using a Weighted Ensemble Approach. 2018, J. Hydrol. Eng., 23(4): 04018004.
Figures 8 and 9 color schemes are confusing. Cold colors should represent a decrease change and warm colors should represent an increase change (or vice-versa). With this colorbar it is visually difficult to distinguish between negative and positive values (e.g. -0.1 and 0.1)
We have redrawn Figure 8 and Figure 9 with different color schemes as the reviewer suggests.
Ln 473-478: There is much literature on how extreme extratropical cyclones may change under future scenarios (e.g. Ulbrich et al. 2013) so further references and information on this could be added to corroborate the discussion of these results. However, some discussion should also include the correlation between obtained results and topography.
We added discussions to lines 125-128, 567-568 and 732-732.
Ln 490-491 – “Nonetheless, these errors are still widely used in the literature, which is one reason why we employ them in our analyses in the previous sections” Please rephrase: what do you mean by “them”? The “errors”?
Corrected in lines 583-584 now as “Nonetheless, the term “return period” is still widely used in the literature, which is one reason that we employ it in our analyses in the previous sections.”
Ln 513 – “etc.” should be avoided
Corrected in line 613 now as “such as the maximum likelihood method and the L-moment method [36, 42].” Thanks.
Ln 548 – “model spread”, as previously stated, “model” and “ensemble” are not synonymous in this study. The text should be revised carefully.
Changed to “ensemble spread”
Figure 10 caption: please indicate what colors represent.
Added as “Figure 10. Northeastern US domain averaged exceedance probabilities (Eqs. 7 and 8) for different relevant historical return periods (related to ) under different scenarios. Results for UWPD (upper; a: RCP4.5; b: RCP8.5) and LOCA (lower; c: RCP4.5; d: RCP8.5). The short vertical lines represent the uncertainties estimated by the ensemble spread. Red represents the results for the historical period (1960-2005), green for the first half of the 21st century (2006-2053), and black for the second half (2054-2100)”. Thanks.
Reference:
Ulbrich et al. (2013) Are Greenhouse Gas Signals of Northern Hemisphere winter extra-tropical cyclone activity dependent on the identification and tracking algorithm? Meteorologische Zeitschrift, Vol. 22, No. 1, 61–68 DOI 10.1127/0941-2948/2013/0420
Reviewer 3 Report
The study investigates the precipitation projection with observation and model data. Overall, the study is properly designed with sound idea, and the results are supportive to the conclusions. The manuscript is well written and descriptive. I suggest acceptance subject to some revisions.
A range of datasets have been processed. In section 2.1., the datasets are briefed. However, it is yet to be detailed about the data quality control, as it is important to subsequent analysis and discussion.
Figure 3 is not properly displayed. The colorbar at end of page seven is confusing.
Lines 530-549. I feel more discussion are necessary to back up the argument that exceedance probability is significant suggesting more frequent extremes events. Maybe showing a table of numbers in addition to Figure 10.
The discussion and conclusions appear to wrap up in a rush. Reword it with more discussions.
Author Response
We thank the reviewers for their valuable comments and suggestions (in red). Here are our point-by-point replies (in blue). We also notice that the layout and line numbers of Word file may change during the online submission process. The line numbers mentioned in this reply may not match exactly that the reviewer receives. Therefore, we uploaded a Word file with all changes tracked.
Reply to Review 3.
The study investigates the precipitation projection with observation and model data. Overall, the study is properly designed with sound idea, and the results are supportive to the conclusions. The manuscript is well written and descriptive. I suggest acceptance subject to some revisions.
A range of datasets have been processed. In section 2.1., the datasets are briefed. However, it is yet to be detailed about the data quality control, as it is important to subsequent analysis and discussion.
We have added more information about the data on lines 119, 122-123, 135-143, 156-165, 171, 221-222, 261-265.
Figure 3 is not properly displayed. The colorbar at end of page seven is confusing.
Corrected. Thanks.
Lines 530-549. I feel more discussion are necessary to back up the argument that exceedance probability is significant suggesting more frequent extremes events. Maybe showing a table of numbers in addition to Figure 10.
We have added Table 2 and related discussions to the paper.
The discussion and conclusions appear to wrap up in a rush. Reword it with more discussions.
We have added more discussions to lines 728-736 and 744-749. Thanks.
Round 2
Reviewer 3 Report
The author addressed me questions. They have provided a revised manuscript that is technically sound and rhetorically well described. It is in a good shape for publication.